



# Potential of $^{14}$C-based versus ΔCO-based ΔffCO$_2$ observations to estimate urban fossil fuel CO$_2$ (ffCO$_2$) emissions

Fabian Maier[1], Christian Rödenbeck[2], Ingeborg Levin[1], Christoph Gerbig[2], Maksym Gachkivskyi[1,3], and Samuel Hammer[1,3]

[1]Institut für Umweltphysik, Heidelberg University, INF 229, 69120 Heidelberg, Germany
[2]Department of Biogeochemical Systems, Max Planck Institute for Biogeochemistry, Jena, Germany
[3]ICOS Central Radiocarbon Laboratory, Heidelberg University, Berliner Straße 53, 69120 Heidelberg, Germany

*Correspondence to*: Fabian Maier (Fabian.Maier@iup.uni-heidelberg.de)

**Abstract.** Atmospheric transport inversions are a powerful tool for independently estimating surface CO$_2$ fluxes from atmospheric CO$_2$ concentration measurements. However, additional tracers are needed to separate the fossil fuel CO$_2$ (ffCO$_2$) emissions from natural CO$_2$ fluxes. In this study we focus on radiocarbon ($^{14}$C), the most direct tracer for ffCO$_2$, and the continuously measured surrogate tracer carbon monoxide (CO), which is co-emitted with ffCO$_2$ during incomplete combustion. In the companion paper by Maier et al. (2023a) we determined for the urban Heidelberg observation site in Southwestern Germany discrete $^{14}$C-based and continuous ΔCO-based estimates of the ffCO$_2$ excess concentration (ΔffCO$_2$) compared to a clean-air reference. Here, we use the CarboScope inversion framework adapted for the urban domain around Heidelberg to assess the potential of both types of ΔffCO$_2$ observations to investigate ffCO$_2$ emissions and their seasonal cycle. We find that discrete $^{14}$C-based ΔffCO$_2$ observations from almost 100 afternoon flask samples collected in the two years 2019 and 2020 are not well suited for estimating robust ffCO$_2$ emissions in the main footprint of this urban area with a very heterogeneous distribution of sources including several point sources. The benefit of the continuous ΔCO-based ΔffCO$_2$ estimates is that they can be averaged to reduce the impact of individual hours with an inadequate model performance. We show that the weekly averaged ΔCO-based ΔffCO$_2$ observations allow for a robust reconstruction of the seasonal cycle of the area source ffCO$_2$ emissions from temporally flat a-priori emissions. In particular, the distinct COVID-19 signal with a steep drop in emissions in spring 2020 is clearly present in these data-driven a-posteriori results. Moreover, our top-down results show a shift in the seasonality of the area source ffCO$_2$ emissions around Heidelberg in 2019 compared to the bottom-up estimates from TNO. This highlights the huge potential of ΔCO-based ΔffCO$_2$ to verify bottom-up ffCO$_2$ emissions at urban stations if the ΔCO/ΔffCO$_2$ ratios can be determined without biases.



## 1 Introduction

The combustion of fossil fuels (ff) like coal, oil and gas is the major reason for the steep increase in the atmospheric $CO_2$ concentration, which causes current global warming. About 70% of the global ff$CO_2$ emissions are released from urban hotspot regions (Duren and Miller, 2012). However, the atmospheric $CO_2$ increase is fortunately strongly weakened, since about half of the human-induced $CO_2$ emissions are currently taken up by the terrestrial biosphere and the oceans in roughly equal shares (Friedlingstein et al., 2022). Indeed, there are large seasonal and inter-annual variations in the natural $CO_2$ sinks and sources

that need to be better understood in order to make predictions about future changes in the carbon cycle owing to increased atmospheric $CO_2$ levels.

The "atmospheric transport inversion" (Newsam and Enting, 1988) is a powerful tool for deducing surface $CO_2$ fluxes from atmospheric $CO_2$ observations. Hence, many studies have applied this top-down approach to constrain natural $CO_2$ fluxes (e.g.,

Rödenbeck et al., 2003; Peylin et al., 2013; Jiang et al., 2016; Rödenbeck et al., 2018; Monteil et al., 2020; Liu et al., 2021).
In these calculations, ff$CO_2$ emissions are typically prescribed using bottom-up information from emission inventories. These bottom-up ff$CO_2$ emission estimates are typically based on national annual activity data that describe the fuel consumption and sector-specific emission factors (Janssens-Maenhout et al., 2019). While annual national total ff$CO_2$ emissions are associated with low uncertainties of typically a few percent for developed countries (Andres et al., 2012), their proxy-based

distribution on individual spatial grid cells and individual months, days or hours can dramatically increase the uncertainties (Super et al., 2020). Therefore, it is essential to have an independent verification of the bottom-up statistics, especially on the relevant urban scales where emission reduction measures are implemented. This is essential on the path to net zero emissions. Furthermore, the seasonal cycle of bottom-up ff$CO_2$ emissions needs to be validated, if they are used in $CO_2$ inversions to deduce biogenic $CO_2$ fluxes that are dominated by a large seasonal cycle.


Atmospheric transport inversions can be used to validate these bottom-up ff$CO_2$ emissions (e.g., Lauvaux et al., 2016; Basu et al., 2020). However, their success relies on the ability of the used observational tracers to separate fossil fuel from natural $CO_2$ contributions (Ciais et al., 2015; Bergamaschi et al., 2018). The most direct tracer for ff$CO_2$ is radiocarbon ($^{14}$C) in $CO_2$. Radiocarbon has a half-life of 5700 years and is therefore no longer present in fossil fuels (Suess, 1955). Thus, the $^{14}$C depletion

in ambient air $CO_2$ compared to a clean-air reference site can directly be used to estimate the recently added ff$CO_2$ excess (Δff$CO_2$) at the observation site (Levin et al., 2003; Turnbull et al., 2006). These Δff$CO_2$ estimates can then be implemented in regional inversions to evaluate bottom-up ff$CO_2$ emissions in the footprints of the observation sites (Graven et al., 2018; Wang et al., 2018). However, a drawback of $^{14}$C-based Δff$CO_2$ estimates is that they have poor temporal and spatial coverage due to the labor-intensive and expensive $^{14}$C sampling and analysis. Therefore, continuously measured atmospheric excess

concentrations of trace gases like CO, which is co-emitted with ff$CO_2$, have been used as alternative proxies for Δff$CO_2$ (e.g., Gamnitzer et al., 2006; Turnbull et al., 2006; Levin and Karstens, 2007; van der Laan et al., 2010; Vogel et al., 2010). However,

to construct a high-resolution $\Delta CO$-based $\Delta ffCO_2$ record requires to correctly determine the $\Delta CO/\Delta ffCO_2$ ratio in the footprint of the observation site. This can indeed be a big challenge: As the $CO/ffCO_2$ emission ratio depends on the combustion efficiency and applied end-of-pipe measures, it is very variable for different emission processes and changes with time due to technological progress (Dellaert et al., 2019).

In the companion paper by Maier et al. (2023a) we calculated a $\Delta CO$-based $\Delta ffCO_2$ record for the urban Heidelberg observation site from the $\Delta CO/\Delta ffCO_2$ ratios of almost 350 $^{14}CO_2$ flask samples collected between 2019 and 2020. By comparing the $\Delta CO$-based $\Delta ffCO_2$ with the $^{14}C$-based $\Delta ffCO_2$ from the flasks we estimated for the hourly $\Delta CO$-based $\Delta ffCO_2$ record an uncertainty of about 4 ppm, which is almost 4 times larger than typical $^{14}C$-based $\Delta ffCO_2$ uncertainties. About half of this uncertainty could be attributed to the spatiotemporal variability of the $\Delta CO/\Delta ffCO_2$ ratios.

The goal of this study is to investigate which type of $\Delta ffCO_2$ observations provides the greater benefit in an atmospheric transport inversion to verify bottom-up $ffCO_2$ emission estimates in an urban region: (1) sparse $^{14}C$-based $\Delta ffCO_2$ observations from flasks with a small uncertainty or (2) $\Delta CO$-based $\Delta ffCO_2$ estimates at high temporal resolution but with an increased uncertainty? For this, we adapt the CarboScope inversion framework (Rödenbeck, 2005) for the highly populated and industrialized Rhine Valley in Southwestern Germany around the Heidelberg observation site. We perform separate inversion runs with the $^{14}C$- and $\Delta CO$-based $\Delta ffCO_2$ observations from Heidelberg. Thereby, we mainly focus on the seasonal cycle in the $ffCO_2$ emissions and investigate which $\Delta ffCO_2$ information is best suited to verify the seasonal cycle of the bottom-up emissions in the main footprint of Heidelberg.

## 2 Methods

### 2.1 Heidelberg observation site

Heidelberg is a medium-sized city with about 160'000 inhabitants, which is part of the Rhine-Neckar metropolitan area with over 2 million people. The Heidelberg observation site is located on the university campus in the northern part of the city. The sampling inlet line is 30 m above ground on the roof of the institute's building. Local $ffCO_2$ emissions originate mainly from traffic and residential heating but there is also a nearby combined heat and power station as well as a large coal-fired power plant and the giant industrial complex from BASF at 15-20 km to the North-West. Due to its location in the Upper Rhine Valley, Heidelberg is frequently influenced by south-westerly air masses, which carry the signals from heterogeneous sources in the Rhine Valley. A more detailed description of the Heidelberg observation site can be found in Levin et al. (2011). The $^{14}C$-based and $\Delta CO$-based $\Delta ffCO_2$ observations from Heidelberg are presented in Sect. 2.2.3.



## 2.2 Inversion setup

The CarboScope inversion algorithm was initially introduced by Rödenbeck et al. (2003) to estimate inter-annual and spatial variability in global $CO_2$ surface-atmosphere fluxes. The algorithm can also be applied to regional inversions (Rödenbeck et al., 2009). In the present study we adapt this inversion modelling framework to estimate $ffCO_2$ surface fluxes in the regional Rhine Valley domain (see Fig. 1) with $\Delta ffCO_2$ observations from the Heidelberg observation site (see Fig. 2). This requires a high-resolution atmospheric transport model and a careful estimation of the lateral $\Delta ffCO_2$ boundary conditions.

The CarboScope inversion system uses Bayesian inference to minimize the deviations between observed and modelled $\Delta ffCO_2$ concentrations by finding the (global) minimum of the cost function (for technical details see Rödenbeck, 2005). This cost function consists of a data constraint and an a-priori constraint, which is needed to regularize the underdetermined problem and to prevent large and unrealistic spatiotemporal $ffCO_2$ flux variabilities (Rödenbeck et al., 2018). The data constraint is weighted by the uncertainties of the transport model and the $\Delta ffCO_2$ observations. Furthermore, the uncertainty applied for the a-priori $ffCO_2$ emissions determines the impact of the a-priori constraint. Overall, the ratio between the model-data uncertainty and the a-priori flux uncertainty controls the strength of the a-priori constraint over the observational constraint (Rödenbeck, 2005; Kountouris et al., 2018; Munassar et al., 2022).

### 2.2.1 Atmospheric transport model

We use the Stochastic Time-Inverted Lagrangian Transport (STILT; Lin et al., 2003; Nehrkorn et al., 2010) model, driven by meteorological fields from the high-resolution Weather Research and Forecasting (WRF) model, to simulate the atmospheric transport in the Rhine Valley domain (see red rectangular in Fig. 1). The WRF meteorological fields have a horizontal resolution of 2 km and are based on hourly 0.25°-resolved European ReAnalysis 5 (ERA5, Hersbach et al., 2020) data from the European Centre for Medium-Range Weather Forecasts (ECMWF). As there are many point source emissions within the Rhine Valley, we apply the STILT volume source influence (VSI) approach introduced by Maier et al. (2022) to model them. This model approach takes into account the effective heights of the point source emissions, which are typically released from elevated chimney stacks. For the area source emissions, we apply the standard approach in STILT, which assumes that all emissions are released from the surface.



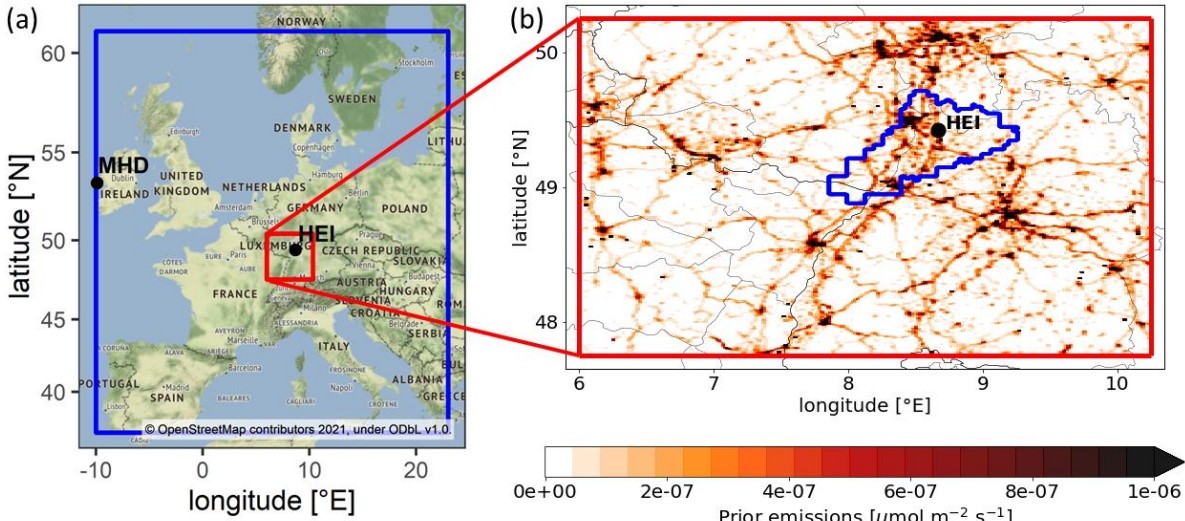

**Figure 1: (a) Map with the Central European STILT domain (blue) and the high-resolution Rhine Valley STILT domain (red). The observation site Heidelberg (HEI) and the marine background site Mace Head (MHD) are indicated. (b) Zoom into the Rhine Valley domain with the mean prior ffCO$_2$ emissions from the TNO inventory for 2019-2020. The blue surrounded region in the zoom shows the "50%-footprint" range, i.e., the area accounting for 50% of the Heidelberg average footprint within the Rhine Valley.**

### 2.2.2 A-priori information

We use the ffCO$_2$ emissions from TNO (Dellaert et al., 2019; Denier van der Gon et al., 2019) with a horizontal resolution of about 1 km (1/60° lon, 1/120° lat) as a-priori estimates for our Rhine Valley inversion. The TNO emission inventory provides annual total ffCO$_2$ emissions for 15 different source sectors as well as sector-specific temporal profiles. In this study, we treat the ffCO$_2$ emissions from the point source dominated "energy production" and "industry" TNO sectors separately due to the following reasons: (1) While the VSI approach (see above) strongly improves the vertical representation of point source emissions in STILT, it still remains difficult to correctly describe the mixing and transport of narrow point source plumes with meteorological fields that have a resolution of 2 km. (2) Due to the elevated release of point source emissions from high stacks, the Heidelberg observation site with an air intake height of only 30m above ground is rarely influenced by distinct emission plumes from nearby point sources (see Maier et al., 2023a). This makes it difficult to evaluate those point source emissions with ΔffCO$_2$ observations from the Heidelberg observation site alone. (3) We expect that the energy and industry emissions in the Rhine Valley are better known in TNO than the more diffuse area source emissions. We thus focus on how well our observations are able to constrain area source emissions in the footprint of the Heidelberg site.

For these reasons, we prescribe the energy and industry emissions in our inversion setup and adjust only the area source emissions in the Rhine Valley, which mainly originate from the heating and traffic sector. We use the sector-specific monthly profiles provided by TNO to calculate from the annual total emissions monthly ffCO$_2$ emissions for the energy and industry sectors and treat them as fixed fluxes in our inversion system. As we aim to investigate the information of the ΔffCO$_2$



observations regarding the seasonal cycle of the area source ffCO$_2$ emissions, we apply temporally constant ("flat") a-priori

ffCO$_2$ emissions for the area sources. For this, we use the (spatially highly resolved) 2-year average TNO area source emissions of the years 2019 and 2020.

### 2.2.3 Observations

In separate inversion runs, we use either the discrete [14]C-based $\Delta$ffCO$_2$ estimates from flasks, collected as integrals over one hour, or the hourly $\Delta$CO-based $\Delta$ffCO$_2$ record from the Heidelberg observation site (see Fig. 2). The companion paper (Maier

et al., 2023a) describes in detail the construction of this continuous $\Delta$CO-based $\Delta$ffCO$_2$ record and gives an estimation of its uncertainty. In short, the $\Delta$CO-based $\Delta$ffCO$_2$ record has been constructed by dividing the observed hourly $\Delta$CO offsets compared to the marine reference site Mace Head (MHD) by an average $\Delta$CO/$\Delta$ffCO$_2$ ratio, which was determined by the $\Delta$CO and [14]C-based $\Delta$ffCO$_2$ observations of almost 350 day- and night-time flask samples collected in 2019 and 2020. In the inversion, however, we only use the afternoon [14]C-based and $\Delta$CO-based $\Delta$ffCO$_2$ observations between 11 and 16 UTC, as

night-time situations are associated with a poorer transport model performance. Note that the hourly-integrated $\Delta$ffCO$_2$ observation e.g. at 11 UTC corresponds to the time period between 11 and 12 UTC.

Furthermore, we apply a 2$\sigma$-selection criterion to the $\Delta$ffCO$_2$ observations as introduced by Rödenbeck et al. (2018). For this, we take the high-resolution annual total ffCO$_2$ emissions from TNO and apply the hourly sector-specific temporal profiles.

These hourly resolved ffCO$_2$ emissions are then transported with the WRF-STILT model to simulate hourly $\Delta$ffCO$_2$ concentrations. The mean difference between the simulated and the $\Delta$CO-based $\Delta$ffCO$_2$ observations is only -0.04ppm during afternoon hours with a standard deviation of 6.76 ppm, which indicates that the model is able to reproduce, on average, the afternoon $\Delta$CO-based $\Delta$ffCO$_2$ observations without a significant mean bias. This directly allows the application of the 2$\sigma$-selection criterion, which means that we only use those $\Delta$ffCO$_2$ observations, whose deviation to the modelled $\Delta$ffCO$_2$ is

smaller than 2 times the standard deviation between observed and modelled $\Delta$ffCO$_2$, i.e. which is within the 2$\sigma$-range. Therewith, we exclude the data outside the 2$\sigma$-range, which obviously cannot be represented with our transport model. Examples of such data are observations during very strong air stagnation events in winter, which are often underestimated in the model, or vice versa, situations when the model overestimates the point source influence at the observation site. Since the inversion system assumes a Gaussian distribution for the model-data mismatch, these extreme outlier events would have a

strong impact on the inversion results (Rödenbeck et al., 2018). Thus, this 2$\sigma$-selection can be seen as an additional regularization for the inversion to avoid using situations with unrealistic model simulations. We apply the 2$\sigma$-selection criterion to both the [14]C-based $\Delta$ffCO$_2$ observations from the afternoon flask samples and the afternoon hours of the $\Delta$CO-based $\Delta$ffCO$_2$ record.





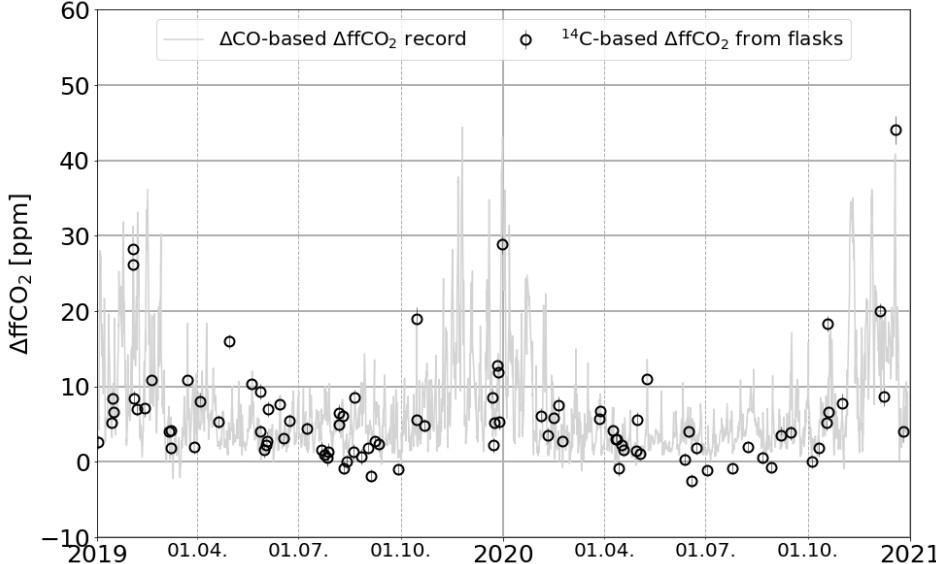

**Figure 2: Afternoon ΔffCO₂ observations from the Heidelberg observation site. The grey curve indicates the ΔCO-based ΔffCO₂ record and the black circles the ¹⁴C-based ΔffCO₂ estimates from flasks. Both, the ¹⁴C-based and ΔCO-based ΔffCO₂ observations are 2σ-selected.**

### 2.2.4 Lateral boundary conditions

We set up the inversion system for the Rhine Valley domain (6.00°E – 10.25°E, 47.75°N – 50.25°N, red rectangular in Fig. 1a) around the Heidelberg observation site and run the inversion for the full two years 2019 and 2020 within this domain. As we calculated the ¹⁴C- and ΔCO-based ΔffCO₂ excess compared to MHD (see Maier et al., 2023a), however, we need to define a suitable ΔffCO₂ background representative for the boundary of the Rhine Valley domain. In the following, we call this the "Rhine Valley ΔffCO₂ background". By definition, we assume that the $\Delta^{14}CO_2$ observations from MHD correspond to ΔffCO₂ = 0 ppm, which might be reasonable since the MHD ¹⁴CO₂ samples were only collected during situations with clean westerly air masses from the Atlantic. Therefore, it seems to be suitable to apply the MHD (ΔffCO₂ = 0 ppm) background to the entire western boundary of the Central European STILT domain (blue rectangular in Fig. 1a). But how representative is this background for the other boundaries of the Central European domain? Maier et al. (2023b) estimated the representativeness bias of the MHD background for the probably most polluted eastern boundary of the Central European domain. They could show that the representativeness bias is on average smaller than 0.1 ppm for an observation site in Central Europe. Therefore, we neglect this bias and assume ΔffCO₂ = 0 ppm also at the non-western boundaries of the Central European domain. To estimate the Rhine Valley ΔffCO₂ background we use a nested STILT model approach with a high (2 km) resolution WRF meteorology in the Rhine Valley domain and a coarser (10 km) WRF resolution in the Central European STILT domain *outside* the Rhine Valley. We use for both domains hourly ffCO₂ emissions from TNO (Dellaert et al., 2019; Denier van der Gon et al., 2019). This nested approach allows us to separate the ffCO₂ contributions from each STILT domain. With this setup we model for the Heidelberg site for each hour during 2019 and 2020 the ΔffCO₂ contributions from the Central European domain




*outside* the Rhine Valley ($\Delta ffCO_{2,CE\text{-}RV}$), which we use as the Rhine Valley background. We then subtract this modelled Rhine Valley $\Delta ffCO_2$ background ($\Delta ffCO_{2,CE\text{-}RV}$) from the estimated $\Delta ffCO_2$ excess compared to MHD ($\Delta ffCO_{2,MHD}$), to obtain the $\Delta ffCO_2$ excess compared to the Rhine Valley boundary ($\Delta ffCO_{2,RV}$):

$$\Delta ffCO_{2,RV} = \Delta ffCO_{2,MHD} - \Delta ffCO_{2,CE\text{-}RV} \tag{1}$$

The $\Delta ffCO_{2,RV}$ excess concentrations compared to the Rhine Valley boundary are then introduced into the inversion system to constrain the $ffCO_2$ emissions within the Rhine Valley.

### 2.2.5 Model-data mismatch

The model-data mismatch is calculated by subtracting the modelled from the observed $\Delta ffCO_{2,RV}$ concentrations. The uncertainties of the $\Delta CO$-based and [14]C-based $\Delta ffCO_2$ observations are estimated to be 3.9 ppm and 1.1 ppm, respectively (see

Maier et al., 2023a). The transport model uncertainty of urban, continental sites like Heidelberg with complex local circulations was assumed to be 5ppm. The quadratically added observational and transport model uncertainties yield the total uncertainty of the model-data mismatch. To account for the temporal correlations of observations that are close together in time, we apply a data density weighting as described in Rödenbeck (2005). It artificially increases the uncertainty of the model-data mismatch, so that all observations within one week lead to the same constraint as a single observation per week. The weighting interval

was set to one week because this is a typical duration of synoptic weather patterns.

### 2.2.6 Degrees of freedom

Since we only use $\Delta ffCO_2$ observations from a single station in the Rhine Valley, we restrict the number of degrees of freedom in our inversion system so that the inverse problem is not too strongly underdetermined. Therefore, we only investigate the area source emissions in the Rhine Valley and prescribe the energy and industry emissions, as mentioned above. Moreover,

the inversion system adjusts only one spatial scaling factor, which increases or decreases the area source emissions in the whole Rhine Valley domain equally. Hence, we expect that the high-resolution TNO inventory is much better at describing the large spatial heterogeneity in the $ffCO_2$ emissions within the Rhine Valley than our top-down approach. As we want to investigate the seasonal cycle of the $ffCO_2$ emissions, additional temporal degrees of freedom are needed. For this, we choose a temporal correlation length of about 4 months ("Filt3T" in CarboScope notation), which should be appropriate to explore

seasonal cycles. Finally, since the Heidelberg observations cannot be used to constrain the emissions in the whole Rhine-Valley domain, we only analyze the a-posteriori area source emissions in the (most constrained) nearfield of the observation site. We define the nearfield of Heidelberg as the area which accounts for 50% of the temporally accumulated footprint in the Rhine Valley domain for the two years 2019 and 2020 (blue surrounded region in Fig. 1b).



# 3 Results

## 3.1 Potential of flask-based ΔffCO₂ estimates to investigate the seasonal cycle in ffCO₂ emissions



**Figure 3: Area source ffCO₂ emissions in the nearfield (blue surrounded area in Fig. 1b) of Heidelberg. Shown are the flat prior emissions (black dashed line), the a-posteriori emissions for different prior uncertainties between 20% and 200% of the flat a-priori emissions (colored solid lines) as well as the bottom-up estimates from TNO (grey line). In panel (a) ¹⁴C-based ΔffCO₂ estimates from 94 2σ-selected afternoon flasks from Heidelberg were used as observational input (cf. Fig. 2). Panel (b) shows the inversion results if the ΔCO-based ΔffCO₂ observations subsampled during the 94 flask sampling hours were used. In the panels (c) and (d) the inversion was constrained with one hourly afternoon (at 13 UTC) ΔCO-based ΔffCO₂ observation every week collected on Tuesday (c) or Friday (d). Panel (e) shows the results if each day at 13 UTC one hypothetical flask is collected. In panel (f), the 7 afternoon flask observations within one week are averaged.**

First, we investigate the potential of flask-based ΔffCO₂ estimates to explore the seasonal cycle of the area source ffCO₂ emissions around the urban Heidelberg observation site. For this we use the average of the TNO area source ffCO₂ emissions





of the two years 2019 and 2020 as a temporally constant prior estimate (see Sect. 2.2.2). To analyze the impact of the
observational constraint on the a-posteriori results, we apply different prior uncertainties, which effectively lead to different
ratios between a-priori and data constraint (see Fig. 3). In a first inversion run (Fig. 3a), we use the $^{14}$C-based $\Delta$ffCO$_2$
observations from the 94 afternoon flasks collected in the two years 2019 and 2020 in Heidelberg. The distribution of the flask
samplings over the two years can be seen in Fig. 2. Due to various reasons (e.g. testing of the flask sampler associated with
frequent changes of the flask sampling strategy) the flasks were not evenly collected and especially the winter 2019/2020 has
only thin flask coverage. The $^{14}$C-based a-posteriori ffCO$_2$ emissions show a clear seasonal cycle for the larger prior
uncertainties, which is fully data-driven. However, large and unrealistic a-posteriori flux variabilities emerge for prior
uncertainties larger than 50% of the flat a-priori emissions. For example, the low flask coverage during the winter period
2019/2020 leads to a huge maximum in the area source ffCO$_2$ emissions in November 2019 when the inversion algorithm tries
to minimize the model-data mismatches of individual flasks. Similarly, the flask samples with vanishing or even negative
$\Delta$ffCO$_2$ estimates in summer 2020 (cf. Fig. 2) cause a strong reduction of the a-posteriori emissions. Therefore, this urban
inversion setup obviously needs a very strong regularization through low prior uncertainties to prevent the fitting of individual
flask observations.

We further investigate whether these overfitting patterns can be attributed to the uneven distribution of the flask samples. For
this, we subsample the continuous $\Delta$CO-based $\Delta$ffCO$_2$ record. In a first step, we use the $\Delta$CO-based $\Delta$ffCO$_2$ observations from
those 94 afternoon hours with flask samplings as observational constraint (Fig. 3b). For the most part, the subsampled $\Delta$CO-
based $\Delta$ffCO$_2$ observations reproduce the a-posteriori results of the $^{14}$C-based $\Delta$ffCO$_2$ estimates. However, there are differences
like the shifted summer minimum in 2019. These differences can be explained by the variability of the $\Delta$CO/$\Delta$ffCO$_2$ ratios that
we fully neglected by using a constant mean ratio for constructing the $\Delta$CO-based $\Delta$ffCO$_2$ record. Thus, when comparing the
results with the TNO seasonality of emissions (grey histogram) it seems obvious that the $^{14}$C-based $\Delta$ffCO$_2$ estimates provide
the more accurate data than the subsampled $\Delta$CO-based $\Delta$ffCO$_2$ record. However, the general similarity between both results
means that we can use the continuous $\Delta$CO-based $\Delta$ffCO$_2$ record to investigate an even data coverage with hypothetical flask
samples collected in Heidelberg. The middle panels in Fig. 3 show the inversion results if the $\Delta$CO-based $\Delta$ffCO$_2$ record is
subsampled for one flask every week on Tuesday (c) or on Friday afternoon (d), respectively. The increased number of evenly
distributed weekly flasks strongly dampens the variability of the a-posteriori results. However, they show large differences
depending on which day of the week the hypothetical afternoon flask was collected. Whereas the Tuesday flasks for example
lead to a quite unrealistic gradual increase in the ffCO$_2$ emissions between January and November 2019, the Friday flasks
show a more realistic seasonal cycle in this year. In contrast, both Tuesday and Friday flasks lead to an unexpected maximum
in summer 2020. This implies that the a-posteriori results are still dependent on the selection of the individual hypothetical
flasks. Therefore, it seems that even a uniform data coverage with a realistic flask sampling frequency of one flask per week
is not sufficient to determine a plausible seasonal cycle of the area source ffCO$_2$ emissions around Heidelberg, as suggested
by the TNO inventory. However, the situation should be better in the case of real, hourly-integrated $^{14}$C flasks that are collected



e.g. once per week, as the average $\Delta CO/\Delta ffCO_2$ ratio used to construct the $\Delta CO$-based $\Delta ffCO_2$ record might be inappropriate for individual hours.


Finally, we investigated the benefit of an extremely high flask sampling frequency with one flask per afternoon (see Fig. 3e). Here, the a-posteriori results seem to approach towards the TNO bottom-up emissions in 2019. However, there are still unexpectedly strong deviations between the top-down and bottom-up estimates in the summer half-year 2020 for increased prior uncertainties. These differences might be caused by individual afternoon hours with a negative model-data mismatch in 280 summer 2020. To reduce the impact of such hours, we perform a separate inversion run where we average the modelled and observational data of all 7 hypothetical afternoon flasks within each week (Fig. 3f). This further reduces the spread of a-posteriori results, particularly in summer 2020, further approaching towards the seasonal amplitude of the bottom-up TNO emissions. Thus, several afternoon flasks per week would be needed so that the influence of individual flasks on the inversion results can be averaged out and a plausible seasonal cycle amplitude in the area source $ffCO_2$ emissions around Heidelberg 285 can be obtained.

Overall, these results show that the a-posteriori estimates are very sensitive to individual flask observations in this target region with very heterogeneously distributed $ffCO_2$ sources. Obviously, the transport model fails to appropriately simulate the $\Delta ffCO_2$ concentrations for individual afternoon hours. This can be explained by remaining shortcomings in the transport model but 290 also by the enormous heterogeneity of the $ffCO_2$ emissions in the footprint of the Heidelberg observation site. As already mentioned in Sect. 2, modelling individual plumes from point source emissions is a particular challenge in this urban region, and e.g. the forward model estimates of point source signals, even with the improved VSI approach seem often incorrect, at least at a temporal resolution of one hour. Therefore, we decided to aggregate our observations over multiple hours, and change the observation operator in the model such that only aggregated model results are compared to aggregated observations. This 295 minimizes the impact of individual hours with poor model performance.

## 3.2 Potential of continuous $\Delta CO$-based $\Delta ffCO_2$ estimates to investigate the seasonal cycle in $ffCO_2$ emissions

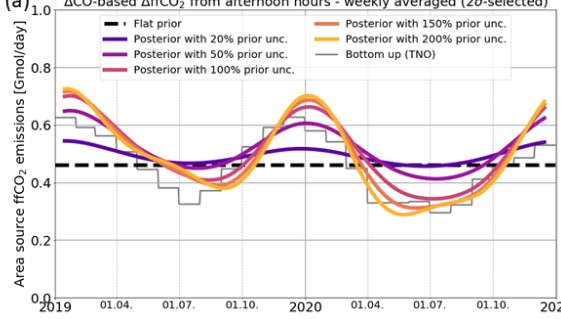
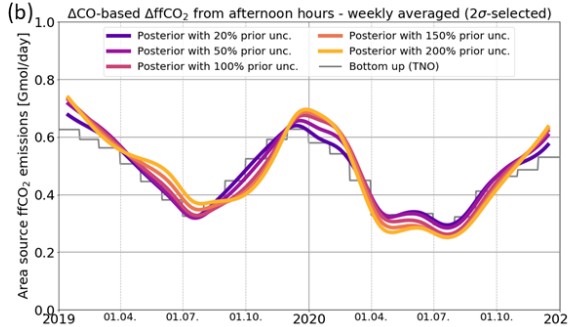



**Figure 4: Area source ffCO$_2$ emissions in the nearfield (blue surrounded area in Fig. 1b) of Heidelberg. In (a) a flat prior (black dashed line) was used for the area source emissions and in (b) the monthly bottom-up estimates from TNO (grey line) were used as**
**a-priori estimate. Shown are the a-posteriori emissions for different prior uncertainties between 20 and 200% (colored solid lines). The inversion was constrained with weekly averages of hourly, 2σ-selected afternoon ΔCO-based ΔffCO$_2$ observations from Heidelberg.**

The big advantage of the continuous ΔCO-based ΔffCO$_2$ record is that it provides a full temporal coverage of the inversion period and can, thus, also be averaged such that the sensitivity of the a-posteriori results on individual (hourly) model-data
mismatches is strongly reduced. In Appendix A1, we investigate if the averaging over the five (11 – 16 UTC) afternoon hours of the ΔCO-based ΔffCO$_2$ record is enough to sufficiently reduce the impact of the point source emissions on the a-posteriori area source emissions (see magenta curves in Fig. A1). For this, we perform in addition to the standard inversion runs with fixed point source emissions further sensitivity runs with adjustable point source emissions. Ideally, the a-posteriori area source emissions are identical for both inversion runs, meaning that the modelling of the well-known point source emissions has no
impact on the area source emissions. However, depending on whether the point source emissions are being adjusted in the inversion framework or not, the a-posteriori area source ffCO$_2$ emissions can differ by more than 100% for individual seasons. Thus, an averaging interval of one afternoon only seems to be too short.

The averaging interval of one week strongly reduces the impact of the point sources on the a-posteriori area source emissions
(see blue curves in Fig. A1). It limits the differences between the a-posteriori area source emissions of the inversion runs with fixed and adjustable point source emissions to below 30% for individual seasons. Averaged over the two years 2019 and 2020, these differences are below 10%. We also tested an extreme averaging interval of one month, which further reduces the difference between the respective a-posteriori area source emissions to below 20% for individual seasons (see pink curves in Fig. A1). However, such a long interval would lead to an averaging of very different meteorological situations, which reduces
the temporal information of the observations. Therefore, we decided to apply in the following an averaging interval of one week, which is the typical length scale of synoptic weather patterns. The difference between the a-posteriori area source emissions of the inversion runs with fixed and adjustable point source emissions can be seen as an uncertainty estimate for the area source emissions due to imperfect point source modelling.

In the following, we use the weekly averaged afternoon ΔCO-based ΔffCO$_2$ observations to investigate the seasonal cycle of the area source ffCO$_2$ emissions around Heidelberg (see Fig. 4a). If the prior uncertainty is chosen large enough, the seasonal cycle amplitude of the a-posteriori estimates agrees with that of the TNO inventory reasonably well. Moreover, the data-driven inversion results distinctly show the effect of the COVID-19 restrictions with lower emissions in 2020 compared to 2019. In Southwestern Germany, the first COVID-19 lockdown started in mid-March 2020. Indeed, the inversion results show at that
time a strong decrease in the area source ffCO$_2$ emissions. In particular, the decline in the a-posteriori ffCO$_2$ emissions is much steeper in spring 2020 compared to spring 2019 and the minimum of the seasonal cycle is flatter in 2020 as it extends over several summer months.





The agreement with the phasing of the seasonal cycle of the TNO inventory seems to be better in 2020 than in 2019. In 2020,
TNO provides country-specific "COVID-19" seasonal cycles, which take into account the timing and the strength of the
respective national restrictions. In Fig. 4 an average over the German and French seasonal cycle is shown, which seems to be
confirmed by our observations. The shown TNO seasonal cycle for 2019 is a general European average estimate that is not
specific for 2019. It assumes minimum emissions in July, whereas our observations show minimal emissions in August and
September. Indeed, this shifted minimum of the seasonal cycle coincides with the summer holidays in Southwestern Germany,
which are from August to mid-September.

We want to further investigate the consistency of the seasonal cycles from the bottom-up and the top-down estimates. For this,
we explore the effect of using the monthly TNO bottom-up seasonal cycle for the a-priori emissions (see Fig. 4b). As expected,
the phase of the a-posteriori seasonal cycle is in agreement with the TNO inventory in 2020. However, in 2019 the a-priori
information pulls the summer emission minimum to July. With weakening regularization of the prior the inversion algorithm
tries to shift the minimum of the a-posteriori seasonal cycle from July towards August and September. Due to the limited
temporal degrees of freedom of the inversion this shifting results in artificially increasing the emissions in May 2019 and
lowering them in October. Hence, these results point to some inconsistencies in the seasonality of the TNO emissions in the
main footprint of the Heidelberg observation site. In fact, a correct phasing of the fossil emissions is essential when prescribed
$ffCO_2$ emissions are used in $CO_2$ model inversions to separate the fossil from the biogenic contribution in atmospheric $CO_2$
observations to constrain $CO_2$ fluxes from the biosphere. Although these biospheric $CO_2$ signals are typically estimated with
observations from sites that are more remote and rural than the urban Heidelberg site, a correct seasonality in the prescribed
$ffCO_2$ emissions still seems to be important when deducing the month-to-month variations in the biospheric $CO_2$ fluxes.

Overall, the (weekly averaged) $\Delta CO$-based $\Delta ffCO_2$ record seems to be well suited to estimate (and verify) the seasonal cycle
of bottom-up $ffCO_2$ emissions in the nearfield of the Heidelberg observation site. This is a very promising result, especially
considering how simple the $\Delta CO$-based $\Delta ffCO_2$ record was constructed. It is based on the average $\Delta CO/\Delta ffCO_2$ ratio estimated
from [14]C measurements on flask samples where a potential seasonal cycle in the $\Delta CO/\Delta ffCO_2$ ratios was fully neglected. In
the following we investigate, among other possible error influences, the effect of a hypothetical seasonal cycle in the
$\Delta CO/\Delta ffCO_2$ ratios on the inversion results.



**3.3 Robustness of the ΔCO-based ΔffCO₂ inversion results**

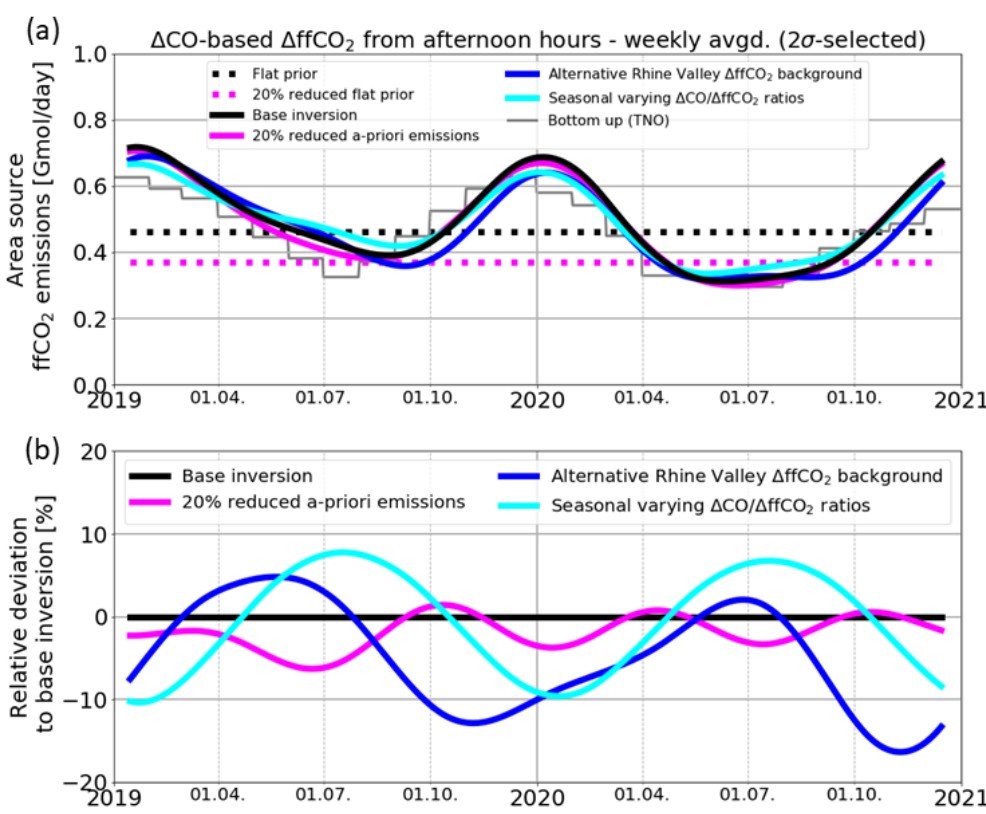

**Figure 5: (a): Area source ffCO₂ emissions in the nearfield (blue surrounded area in Fig. 1b) of Heidelberg for different sensitivity runs. Shown are the a-posteriori results for 20% reduced flat a-priori emissions (solid magenta line), for an alternative Rhine Valley ΔffCO₂ background modelled with EDGAR emissions (blue) and for an assumed seasonal cycle in the ΔCO/ΔffCO₂ ratios (cyan, see Fig. A2). As a reference, the a-posteriori result of the base inversion from Fig. 4a is shown in black. All a-posteriori results correspond to a 150% prior uncertainty. The dotted lines indicate the flat prior emissions (black) and the by 20% reduced prior emissions (magenta). (b): Relative deviations between the different a-posteriori area source emissions of the sensitivity runs and the base inversion in %.**

In the following we want to investigate the robustness of the (weekly averaged) ΔCO-based ΔffCO₂ inversion results. For this, we (1) reduce the flat prior emissions by 20%, (2) assume a seasonal cycle in the ΔCO/ΔffCO₂ ratios, and (3) apply an alternative Rhine Valley ΔffCO₂ background. Figure 5 shows the respective a-posteriori results for a 150% prior uncertainty, which constitutes enough weighting on the (weekly averaged) ΔCO-based ΔffCO₂ observations to reconstruct the seasonal cycle from the flat a-priori area source ffCO₂ emissions (see Fig. 4a).


First, if the a-priori area source ffCO₂ emissions in the Rhine Valley domain are equally reduced by 20% (see dotted magenta curve in Fig. 5a), the ΔCO-based ΔffCO₂ inversion manages to compensate for almost all of this bias (compare the magenta curves with the black curves in Fig. 5). The deviations between the a-posteriori emissions of the inversion runs with perturbed



and unperturbed flat prior emissions is typically below 5% for all seasons. Accordingly, the a-posteriori seasonal cycle of the
ffCO$_2$ emissions is hardly affected by a potential bias in the flat prior emissions. The deviations between the annual totals of
the a-posteriori estimates of the perturbed and unperturbed prior inversion runs is only 2% for both years. This means that on
an annual scale about 90% of this 20%-bias in the perturbed flat prior could be corrected for.

With the second sensitivity test, we want to investigate the effect of the $\Delta$CO/$\Delta$ffCO$_2$ ratios used to construct the $\Delta$CO-based
$\Delta$ffCO$_2$ record. For our base inversion, the $\Delta$CO-based $\Delta$ffCO$_2$ record was constructed by using the average $\Delta$CO/$\Delta$ffCO$_2$ ratio
of 8.44 ppb/ppm, which was calculated from all flask samples collected in 2019 and 2020. However, as discussed in Maier et
al. (2023a), the ratio during summer with lower signals is hard to determine and thus less constrained. The question is thus:
How would our inversion results change if the $\Delta$CO/$\Delta$ffCO$_2$ ratios would have a (small) seasonal cycle? For this, we assume
a seasonal cycle in the ratios for the two years with 5% lower ratios in summer and correspondingly 5% larger ratios in winter,
so that the two-year mean is still 8.44 ppb/ppm (see Fig. A2). Notice, that we use the ratios to calculate the $\Delta$ffCO$_{2,MHD}$ excess
compared to the MHD background site and then subtract the modelled Rhine Valley $\Delta$ffCO$_{2,CE-RV}$ background to get the
$\Delta$ffCO$_{2,RV}$ observations for our Rhine Valley inversion (see Eq. 1). This effectively results in summer and winter $\Delta$ffCO$_2$
concentrations being more than 5% larger and lower, respectively, than the $\Delta$ffCO$_2$ concentrations based on the average ratio.
Obviously, this leads to larger a-posteriori emissions (cyan curve in Fig. 5) during summer and lower emissions in winter
compared to the base inversion results. The largest seasonal deviations to the base inversion a-posteriori emissions are 10%.
Since by construction the mean of the seasonally varying ratios corresponds to the average ratio used for the base inversion,
the effect on the annual totals of the a-posteriori ffCO$_2$ emissions is neglectable.

Finally, we investigate the impact of the lateral $\Delta$ffCO$_2$ boundaries on the area source ffCO$_2$ emissions estimates. For our base
inversion, we used the high-resolution TNO emission inventory and WRF-STILT to model for the Heidelberg observation site
the $\Delta$ffCO$_2$ contributions from the European STILT domain outside the Rhine Valley (see Sect. 2.2.4). For the following
sensitivity run (blue curve in Fig. 5), we model the Rhine Valley $\Delta$ffCO$_2$ background with ffCO$_2$ emissions based on the
Emissions Database for Global Atmospheric Research (EDGAR, Janssens-Maenhout et al., 2019) and use the coarser ECMWF
meteorology in STILT. The application of this alternative Rhine Valley $\Delta$ffCO$_2$ background leads to more than 10% lower
emissions in the autumn of both years, which can be explained by strong deviations between the weekly averages of the two
modelled background concentrations during these periods (see Fig. A3). Thus, the Rhine Valley background affects the
seasonal cycle of the area source ffCO$_2$ emissions. During summer, the deviations to the base inversion results are below 5%.
The annual totals of the area source ffCO$_2$ emissions around Heidelberg are 3% and 7% lower in the years 2019 and 2020,
respectively, if the alternative Rhine Valley $\Delta$ffCO$_2$ background is used.





## 4 Discussion

In the present study we investigate the potential of [14]C-based and $\Delta$CO-based $\Delta ffCO_2$ observations to evaluate the $ffCO_2$ emissions and their seasonal cycle in an urban region around the Heidelberg observation site. This urban area is characterized by a complex topography and a large spatial heterogeneity in the $ffCO_2$ sources, including several nearby point sources. Thus, deficits in the transport model and inaccuracies in the driving meteorology strongly impact the model-data mismatch at the observation site, which will be minimized by the inversion algorithm. We focus on the estimation of the $ffCO_2$ emissions from area sources, since the observations from the Heidelberg site with an air intake height of 30 m above the ground are not suitable to constrain the emissions of nearby point sources with elevated stack heights. Indeed, the analysis of the $\Delta CO/\Delta ffCO_2$ ratios in Maier et al. (2023a) showed that the Heidelberg observation site is hardly influenced by pure point source emission plumes. Moreover, we expect that point source emissions can be better quantified a-priori from bottom up compared to area source emissions. Therefore, we prescribe the well-known point source $ffCO_2$ emissions in the inversion setup and only adjust the area source emissions in the Rhine Valley domain.

### 4.1 Can flask-based $\Delta ffCO_2$ observations be used to predict the seasonal cycle of $ffCO_2$ emissions at an urban site?

To investigate the potential of $\Delta ffCO_2$ observations to predict the seasonal cycle of the area source $ffCO_2$ emissions around Heidelberg, we applied temporally constant (flat) a-priori $ffCO_2$ emissions in our inversion system, such that all seasonal information comes from the atmospheric data. We could show that [14]C-based $\Delta ffCO_2$ observations from almost 100 hourly flask samples collected in the two years 2019 and 2020 are not sufficient to reconstruct a robust seasonal cycle from the flat a-priori estimate. As the Bayesian inversion setup assumes a Gaussian distribution for the model-data mismatch, the inversion algorithm tries to primarily reduce the largest model-data differences. Therefore, we applied a $2\sigma$ selection to exclude the flask events with the largest model-data mismatches and thus worse model performances. However, the a-posteriori $ffCO_2$ emissions are still very sensitive to individual flask observations. Therefore, a strong regularization through small a-priori uncertainties (i.e. < 50% prior uncertainty, Fig. 3a) is needed to avoid large overfitting patterns in the inversion results.

By subsampling the $\Delta CO$-based $\Delta ffCO_2$ record, we further investigate the potential of a uniform data coverage with one hypothetical afternoon flask per week to reliably estimate the seasonal cycle in the area source emissions. Indeed, several afternoon flask samples per week are needed, as well as an averaging of the flask observations within one week so that the overfitting of individual flask data is reduced. However, the situation should be better for real, e.g. sub-weekly, [14]C flasks compared to the subsampled $\Delta CO$-based $\Delta ffCO_2$ record; as the applied average $\Delta CO/\Delta ffCO_2$ ratio may be inappropriate for individual hours, this could amplify the sensitivity to the individual hypothetical flasks. For the Heidelberg observation site where the flask sampling during our study period was rather irregularly, we used the $\Delta CO/\Delta ffCO_2$ flask ratios to construct a continuous $\Delta CO$-based $\Delta ffCO_2$ record (see Maier et al., 2023a) that, due to its high frequency could then be averaged in the inversion framework to explore the seasonal cycle of the $ffCO_2$ emissions.



## 4.2 What is an appropriate averaging interval for urban observations?

The main advantage of the $\Delta CO$-based $\Delta ffCO_2$ record is its continuous data coverage that allows an averaging so that the influence of individual hours with poor model performance on the inversion results is strongly reduced. In this urban region, this is especially necessary because of the shortcomings in the STILT model and its driving meteorology to describe the transport and mixing of nearby point source emissions. Imagine that the plume of a point source arrives a few hours earlier or later at the observation site than simulated by STILT. In such cases, averaging is inevitable to prevent a wrong adjustment of the $ffCO_2$ emissions. Moreover, the STILT-VSI approach itself has its deficits as it assumes mean effective emission height profiles for all meteorological situations and ignores the stack heights of individual power plants. Furthermore, the VSI approach still relies on a correct vertical mixing in STILT. Whereas in Maier et al. (2022) we could show that the VSI approach strongly improves the agreement between modelled and observed $\Delta ffCO_2$ concentrations from two-week integrated samples, it thus still may overestimate the point source contributions for individual hours. Therefore, an averaging of the observations is very helpful when a transport model like STILT is used to describe the transport and mixing of nearby point source emissions.

In Appendix A1, we investigate how to appropriately average the observational data. Ideally, the a-posteriori area source $ffCO_2$ emissions are independent of a wrong modelling of the point source emissions. Thus, they should not be affected by whether the a-priori point source emissions are fixed or adjustable in the inversion framework. We showed that an averaging interval of one week limits the differences between the a-posteriori area source $ffCO_2$ emissions of the inversion runs with fixed and adjustable point source emissions, respectively, to below 30% for all seasons. This deviation can be used as a measure for the uncertainty of the a-posteriori area source $ffCO_2$ emissions that is induced by an inadequate modelling of the point source emissions. A longer, e.g. monthly, averaging interval further reduces this difference, but comes along with an averaging over very different meteorological situations and thus reduces the spatiotemporal information comprised in the observations. This might be especially important if there are several observation sites, and the inversion system optimizes the $\Delta ffCO_2$ gradients between these different stations. The averaging interval of a week corresponds to the typical length scale of synoptic weather patterns. Therefore, a certain correlation between the model-data-mismatch uncertainty within one week has anyhow to be considered in the inversion and the weekly averaging should, thus, not destroy too much information. In this study, we thus applied an averaging interval of one week as a compromise between reducing the impact of hours with an inadequate model performance and using as much observational information as possible.

## 4.3 What is the potential of $\Delta CO$-based $\Delta ffCO_2$ to estimate the seasonal cycle in urban $ffCO_2$ emissions?

The potential of weekly averaged $\Delta CO$-based $\Delta ffCO_2$ observations to explore the seasonal cycle of the $ffCO_2$ emissions in an urban region is very promising. In Heidelberg, we could reliably reconstruct the seasonal cycle from flat a-priori area source $ffCO_2$ emissions with the $\Delta CO$-based $\Delta ffCO_2$ observations for increased prior uncertainties. We further could detect the





COVID-19 signal in 2020, which is characterized by lower emissions compared to 2019 and a very steep decline in the emissions in spring 2020. In this latter year, the a-posteriori seasonal cycle agrees very well with the bottom-up seasonal cycle from TNO, where the timing of the COVID-19 restrictions has explicitly been considered. For 2019, TNO only provides a non-year-specific European average seasonal cycle, which has its annual minimum in July. In contrast, our $\Delta CO$-based $\Delta ffCO_2$ observations suggest the 2019 minimum of the (restricted) Rhine Valley area source $ffCO_2$ emissions to be in August and September, when local summer holidays take place in that part of Germany. Even when we apply the bottom-up seasonal cycles from TNO to the flat a-priori $ffCO_2$ emissions the inversion system still tries to shift the minimum in 2019 from July into September. However, due to the limited temporal degrees of freedom in our inversion system, this comes along with artificially increased or decreased emissions in May and October 2019, respectively. Thus, this result of the Heidelberg inversion points to some inconsistencies in the seasonality of TNO emissions in the footprint of the station. A correct phasing of the fossil emissions is essential when prescribed $ffCO_2$ emissions and associated forward modelling results are used in atmospheric transport inversions to constrain the $CO_2$ fluxes from the biosphere.

In contrast to the inversion with flask-$^{14}$C-based $\Delta ffCO_2$ observations, the $\Delta CO$-based $\Delta ffCO_2$ inversion allows a weakening of the regularization strength without generating unrealistic variabilities in the seasonal cycle of the $ffCO_2$ emissions. This implies that the a-posteriori results are less dependent on a potential bias in the a-priori emissions. Indeed, a sensitivity run with a 20% reduced flat prior estimate for the area source $ffCO_2$ emissions leads for sufficiently large prior uncertainties to similar results as the base inversion run with unperturbed prior estimate. Thus, the $\Delta CO$-based $\Delta ffCO_2$ inversion is able to simultaneously reconstruct the seasonal cycle from a flat prior and correct a potential bias in the a-priori emissions.

However, the $\Delta CO$-based $\Delta ffCO_2$ inversion results strongly depend on a potential bias in the $\Delta CO/\Delta ffCO_2$ ratios that are applied to calculate the $\Delta ffCO_2$ estimates. Since there is no evidence for a strong seasonal cycle in the $\Delta CO/\Delta ffCO_2$ ratios at the Heidelberg observation site, we used a constant average $\Delta CO/\Delta ffCO_2$ ratio to calculate the $\Delta CO$-based $\Delta ffCO_2$ record for the two years 2019 and 2020 (see Maier et al., 2023a). But due to the low signals and the weak correlation between $\Delta CO$ and $\Delta ffCO_2$ during summer, it is hard to determine separate summer ratios. Nevertheless, our results indicate that there might be a small seasonal cycle on the order of 5% in the ratio. In this study, we could show that a hypothetical seasonal cycle with 5% lower and 5% larger ratios in summer and winter, respectively, would lead to changes in the area source $ffCO_2$ emissions of up to 10% for individual seasons. This emphasizes the importance of a thorough determination of the $\Delta CO/\Delta ffCO_2$ ratios to prevent biases in estimates of total fluxes *and* the seasonal cycle of the $ffCO_2$ emissions.

Indeed, we are currently in a kind of a fortunate situation in Heidelberg, since the emission ratios of the traffic and heating sectors seem to be quite similar in the main footprint of the station (see Maier et al., 2023a). Hence, despite the varying share of traffic and heating over the course of a year, this simply allowed the usage of a constant average flask-based $\Delta CO/\Delta ffCO_2$



ratio for constructing the $\Delta CO$-based $\Delta ffCO_2$ record. Of course, it is much more challenging to determine continuous $\Delta CO$-based $\Delta ffCO_2$ estimates for stations where the $\Delta CO/\Delta ffCO_2$ ratios show large seasonal or even diurnal variabilities.

A common challenge in regional inversions is the determination of the lateral boundary conditions (Munassar et al., 2023). In this study, we used two different emission inventories and meteorological fields to estimate the $\Delta ffCO_2$ background for the Rhine Valley domain by modelling the contributions from the Central European $ffCO_2$ emissions outside the Rhine Valley. For individual seasons the a-posteriori area source $ffCO_2$ emissions around Heidelberg can differ by more than 10%. This highlights the strong need for appropriate boundary conditions. In Europe, the Integrated Carbon Observation System (ICOS,

Heiskanen et al., 2022) provides high-quality atmospheric in-situ data from a network of tall-tower stations that cover a large part of the European continent. These observations may help to verify the $ffCO_2$ emissions in Europe. Then, the optimized European $ffCO_2$ emissions could be used to estimate more reliably the $\Delta ffCO_2$ background for the Rhine Valley domain.

Overall, our results demonstrate that the weekly averaged $\Delta CO$-based $\Delta ffCO_2$ observations are currently well suited to

investigate the amplitude and the phasing of the seasonal cycle of the area source $ffCO_2$ emissions in the main footprint of the Heidelberg observation site. The different sensitivity runs suggest that $\Delta CO$-based $\Delta ffCO_2$ allows a reconstruction of this seasonal cycle from temporally constant a-priori estimates with an uncertainty of below ca. 30% for all seasons. Thus, one may recommend applying this $\Delta CO$-based $\Delta ffCO_2$ inversion at further urban sites with a strong heterogeneity in the local $ffCO_2$ sources if the $\Delta CO/\Delta ffCO_2$ ratios can be determined accurately.


Finally, the $\Delta CO$-based $\Delta ffCO_2$ inversion can be seen as a simplification of a multi-species inversion, which is based on collocated $CO_2$ and $CO$ observations. Such a multi-species inversion exploits the fact that the collocated $CO_2$ and $CO$ observations are affected by the same atmospheric transport and that these two species have partially overlapping emission patterns (Boschetti et al., 2018). Boschetti et al. (2018) show that the consideration of these inter-species correlations leads to

a reduction in the respective a-posteriori uncertainties of the $ffCO_2$ (and $CO$) emissions. While our $\Delta CO$-based $\Delta ffCO_2$ inversion assumes a constant but observation-based $\Delta CO/\Delta ffCO_2$ ratio, the multi-species inversion intrinsically considers the spatiotemporal variability of the ratios. However, this requires reliable a-priori estimates of the $CO/ffCO_2$ emission ratios and their uncertainties, as well as neglectable non-fossil $CO$ sources and sinks.

## 5 Conclusions

This study illustrates the strong potential of continuous $\Delta CO$-based $\Delta ffCO_2$ observations to determine the seasonal cycle of $ffCO_2$ emissions in an urban region with highly heterogeneous $ffCO_2$ sources in its vicinity. The ability of averaging and thus reducing the influence of individual hours with an inadequate model performance makes $\Delta CO$-based $\Delta ffCO_2$, despite its larger



uncertainty, currently a better tracer than discrete [14]C-based $\Delta$ffCO$_2$ from weekly flasks for estimating the seasonal cycle of the area source ffCO$_2$ emissions in the Upper Rhine Valley around the urban Heidelberg observation site.


For our study, we set up the CarboScope inversion system in the Rhine Valley. It is based on the high-resolution WRF-STILT model and uses the STILT volume source influence (VSI) approach developed in Maier et al. (2022) to represent the emission heights of point sources. However, despite the high-resolution WRF meteorology and the improved STILT-VSI approach, almost 100 [14]C-based $\Delta$ffCO$_2$ estimates from flasks collected in Heidelberg during the two years 2019 and 2020 are insufficient

to robustly estimate month-to-month variations of the area source ffCO$_2$ emissions in the main footprint of the site. Indeed, it seems that several flasks per week would be needed, so that the flask observations within one week could be averaged and overfitting of individual flask observations be prevented.

Due to the fortunate circumstance of currently having similar heating and traffic emission ratios in the main footprint of

Heidelberg, we could use the average [14]C-based $\Delta$CO/$\Delta$ffCO$_2$ ratio from the flasks to construct a continuous $\Delta$CO-based $\Delta$ffCO$_2$ record (see Maier et al., 2023a). The weekly averaging of this $\Delta$CO-based $\Delta$ffCO$_2$ record strongly reduces the impact of hours with an inadequate modelling on the a-posteriori ffCO$_2$ emissions. In fact, the weekly averaged $\Delta$CO-based $\Delta$ffCO$_2$ observations can robustly reconstruct the amplitude and the phasing of the seasonal cycle of the ffCO$_2$ area source emissions even from temporally constant a-priori emissions. In particular, the observational data clearly contain the distinct COVID-19

signal in 2020, which is characterized by overall lower emissions compared to 2019 and a steep drop in emissions in spring 2020 with the onset of the restrictions. Moreover, the comparison with the bottom-up emissions from TNO points to some inconsistencies in the TNO seasonality of the area source ffCO$_2$ emissions in the footprint of Heidelberg in 2019.

Overall, our sensitivity runs suggest that we can reconstruct the seasonal cycle of the ffCO$_2$ area source emissions around

Heidelberg with an uncertainty of below ca. 30%. Therefore, one may recommend applying the $\Delta$CO-based $\Delta$ffCO$_2$ inversion at further urban sites with heterogeneous ffCO$_2$ sources, if the $\Delta$CO/$\Delta$ffCO$_2$ ratios can be estimated accurately. If ratios from bottom-up inventories are not trusted or the urban region is influenced by CO emissions from the biosphere, the ratios are most reliably calculated from [14]C flasks. Then, at least some of the summer [14]C flasks should be collected during situations with significant CO and ffCO$_2$ signals, so that a possible seasonal cycle in the $\Delta$CO/$\Delta$ffCO$_2$ ratios could be identified. At remote

sites, such as at several ICOS atmosphere stations, with low ffCO$_2$ signals and predominant biosphere influence the calculation of $\Delta$CO/$\Delta$ffCO$_2$ ratios and the construction of a bias-free $\Delta$CO-based $\Delta$ffCO$_2$ record might be more challenging than at an urban site. However, the model performance is expected to be better at remote sites with a typically higher air intake above the ground and a much lower heterogeneity in the surrounding ffCO$_2$ sources with minor influences from nearby point sources. Consequently, the outcome of our urban study cannot directly be transferred to remote sites; further studies are needed to

investigate the potential of [14]C-based versus $\Delta$CO-based $\Delta$ffCO$_2$ to estimate ffCO$_2$ emissions at such sites.





# Appendix

## A1. Impact of point sources on the a-posteriori area source ffCO$_2$ emissions

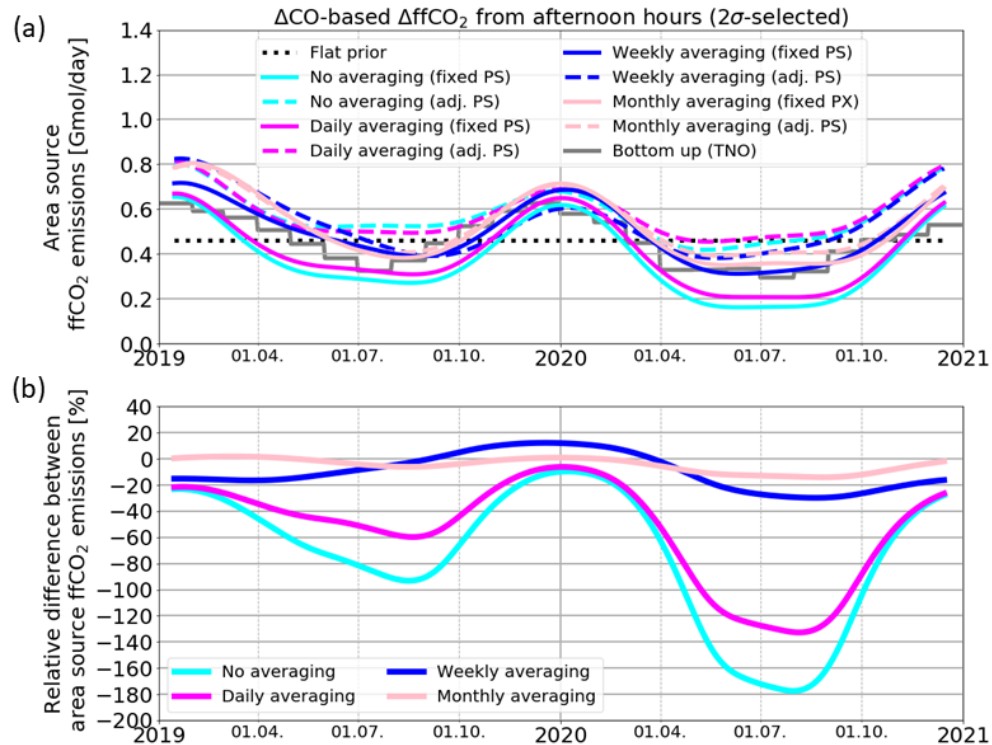

**Figure A1: (a): Area source ffCO$_2$ emissions in the nearfield (blue surrounded area in Fig. 1b) of Heidelberg. Shown are the results**
**of the ΔCO-based ΔffCO$_2$ inversion with fixed point sources (solid lines, "fixed PS") and adjustable point sources (dashed lines, "adj. PS") for different averaging intervals ranging from no averaging at all (cyan) to daily averaging of the five hours of each afternoon (magenta) and weekly (blue) and monthly (pink) averaging. All a-posteriori results correspond to a 150% prior uncertainty. The flat a-priori emissions and the bottom-up emissions are shown as a reference in black and grey, respectively. (b): Relative differences (fixed PS minus adj. PS) between the a-posteriori area source ffCO$_2$ emissions of the inversion runs with adjustable and fixed point**
**source emissions in %.**

To investigate the influence of inadequate point source modelling on the a-posteriori area source ffCO$_2$ emissions, we use two

different ΔCO-based ΔffCO2 inversion setups: (1) an inversion with fixed point source emissions ("INV_fix") and (2) an

inversion with adjustable point source emissions ("INV_adj"). The first inversion setup corresponds to the inversion described

in Sect. 2. It optimizes the flat a-priori area source emissions by using fixed monthly point source emissions. The second

inversion setup optimizes both, the flat a-priori area source emissions, and the monthly a-priori point source emissions.

Thereby, the point source emissions from the energy production and the industry sector, respectively, get the same temporal

(i.e. "Filt3T" in CarboScope notation, see Sect. 2.2.6) and spatial (i.e. one spatial scaling factor) degrees of freedom like the

area source emissions. Ideally, both inversion setups should lead to the same a-posteriori area source emissions, meaning that

the modelling of the well-known point source emissions has no influence on the area source emission estimates. Obviously,





this is not the case. If the model-data mismatches of the individual afternoon hours of the ΔCO-based ΔffCO2 record are not
averaged, the INV_fix inversion leads to much lower area source emissions estimates than the INV_adj inversion (see cyan
curves in Fig. A1). For individual seasons, e.g. in summer 2020, the differences are larger than 150%. Thus, the INV_fix
inversion tends to decrease the area source emissions to compensate for an inadequate modelling of the (fixed) point source
emissions. This shows that even with the VSI approach the model seems to overestimate the contributions from point sources
at the Heidelberg observation site for individual hours.

The averaging over one afternoon (magenta curve in Fig. A1) leads only to minor improvements; there are still deviations
larger than 100% in summer 2020. In contrast, the averaging interval of one week (blue curve) limits the largest deviations in
summer 2020 to below 30%. Averaged over the two years 2019 and 2020, these deviations between the INV_fix and INV_adj
a-posteriori area source emissions are even less than 10%. A monthly averaging interval (pink curve) further reduces the
deviations to below 20% in summer 2020.

## A2. Hypothetical seasonal cycle in the ΔCO/ΔffCO₂ ratios

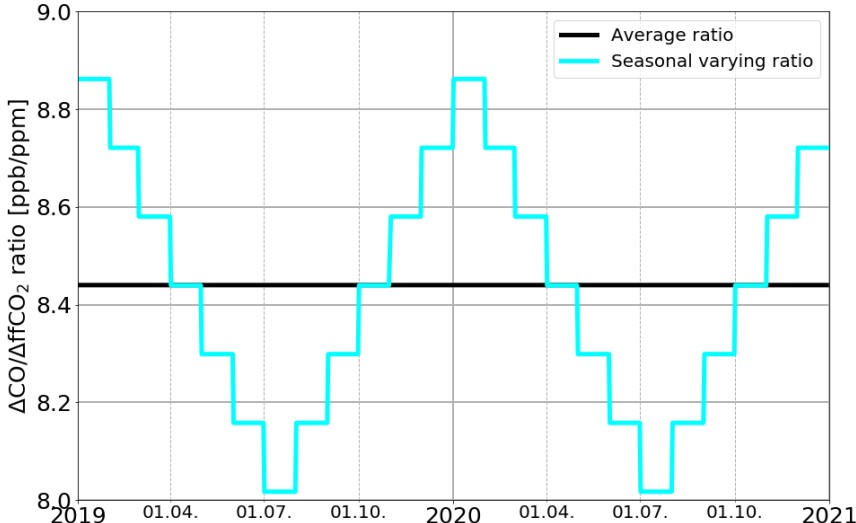

**Figure A2: Average ΔCO/ΔffCO₂ ratio (black) and hypothetical seasonal varying ratio (cyan) used to construct the ΔCO-based**
**ΔffCO₂ record for the base inversion (Fig. 4) and the sensitivity inversion run (cyan curve in Fig. 5), respectively.**



## A3. Comparison between two modelled Rhine Valley $\Delta ffCO_2$ backgrounds

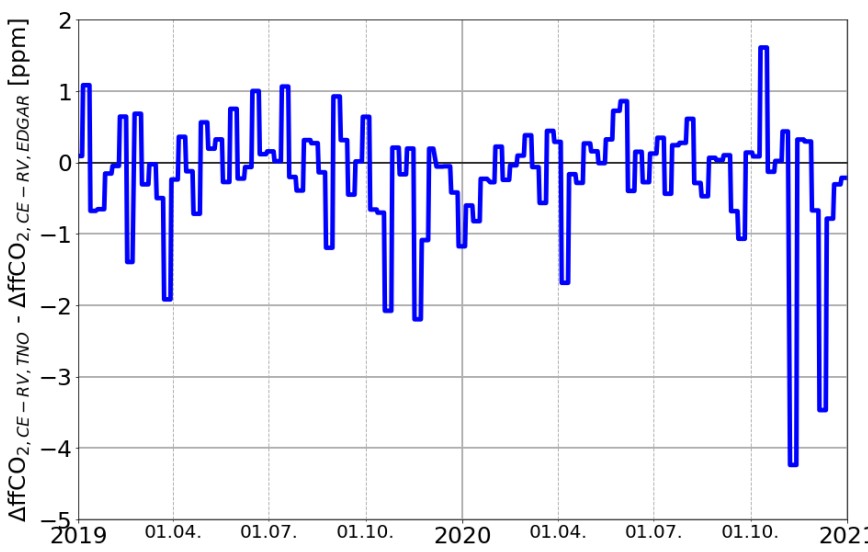

**Figure A3: Difference between the Rhine Valley background modelled with TNO emissions ($\Delta ffCO_{2,CE-RV,TNO}$) and the Rhine Valley background modelled with EDGAR emissions ($\Delta ffCO_{2,CE-RV,EDGAR}$). Shown are weekly averages for afternoon situations.**





**Data availability**

We used the $^{14}$C- and $\Delta$CO-based $\Delta$ffCO$_2$ data published with Maier et al. (2023a).

**Author contribution**

FM designed the study together with all co-authors. FM performed the inverse modelling and processed the inversion results. CR helped with applying the CarboScope inversion framework for the Rhine Valley domain. CG modelled the alternative Rhine Valley $\Delta$ffCO$_2$ background with emissions based on EDGAR. The various inversion results were discussed by all authors. FM wrote the manuscript with help of all co-authors.

**Competing interests**

Some authors are members of the editorial board of ACP. The peer-review process was guided by an independent editor, and the authors have also no other competing interests to declare.

**Acknowledgement**

We are grateful to the staff of TNO at the Department of Climate, Air and Sustainability in Utrecht for providing the emission inventory as well as to Julia Marshall and Michał Gałkowski for computing and processing the high-resolution WRF meteorology in the Rhine Valley.

**Financial support**

This research has been supported by the German Weather Service (DWD), the ICOS Research Infrastructure and VERIFY (grant no. 776810, Horizon 2020 Framework). The ICOS Central Radiocarbon Laboratory is funded by the German Federal Ministry of Transport and Digital Infrastructure.





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
