# Peer review of "Potential of 14C-based versus $\triangle$ CO-based $\triangle$ ffCO2 observations to estimate urban fossil fuel CO2 (ffCO2) emissions"

_EGUsphere, 2023_

## Referee Comment (RC2)

**Potential of [14]C-based versus ΔCO-based ΔffCO₂ observations to estimate urban fossil fuel CO₂ (ffCO₂) emissions**

Fabian Maier[1], Christian Rödenbeck[2], Ingeborg Levin[1], Christoph Gerbig[2], Maksym Gachkivskyi[1,3], and Samuel Hammer[1,3]

[1]Institut für Umweltphysik, Heidelberg University, INF 229, 69120 Heidelberg, Germany
[2]Department of Biogeochemical Systems, Max Planck Institute for Biogeochemistry, Jena, Germany
[3]ICOS Central Radiocarbon Laboratory, Heidelberg University, Berliner Straße 53, 69120 Heidelberg, Germany

*Correspondence to*: Fabian Maier (Fabian.Maier@iup.uni-heidelberg.de)

**Abstract.** Atmospheric transport inversions are a powerful tool for independently estimating surface $CO_2$ fluxes from atmospheric $CO_2$ concentration measurements. However, additional tracers are needed to separate the fossil fuel $CO_2$ (ff$CO_2$) emissions from natural $CO_2$ fluxes. In this study we focus on radiocarbon ([14]C), the most direct tracer for ff$CO_2$, and the continuously measured surrogate tracer carbon monoxide (CO), which is co-emitted with ff$CO_2$ during incomplete combustion. In the companion paper by Maier et al. (2023a) we determined for the urban Heidelberg observation site in Southwestern Germany discrete [14]C-based and continuous ΔCO-based estimates of the ff$CO_2$ excess concentration (Δff$CO_2$) compared to a clean-air reference. Here, we use the CarboScope inversion framework adapted for the urban domain around Heidelberg to assess the potential of both types of Δff$CO_2$ observations to investigate ff$CO_2$ emissions and their seasonal cycle. We find that discrete [14]C-based Δff$CO_2$ observations from almost 100 afternoon flask samples collected in the two years 2019 and 2020 are not well suited for estimating robust ff$CO_2$ emissions in the main footprint of this urban area with a very heterogeneous distribution of sources including several point sources. The benefit of the continuous ΔCO-based Δff$CO_2$ estimates is that they can be averaged to reduce the impact of individual hours with an inadequate model performance. We show that the weekly averaged ΔCO-based Δff$CO_2$ observations allow for a robust reconstruction of the seasonal cycle of the area source ff$CO_2$ emissions from temporally flat a-priori emissions. In particular, the distinct COVID-19 signal with a steep drop in emissions in spring 2020 is clearly present in these data-driven a-posteriori results. Moreover, our top-down results show a shift in the seasonality of the area source ff$CO_2$ emissions around Heidelberg in 2019 compared to the bottom-up estimates from TNO. This highlights the huge potential of ΔCO-based Δff$CO_2$ to verify bottom-up ff$CO_2$ emissions at urban stations if the ΔCO/Δff$CO_2$ ratios can be determined without biases.

[revised manuscript text omitted]

---

## Author Comment (AC1)

We want to thank Jocelyn Turnbull for the review of our manuscript and the helpful suggestions for improvements. Our replies are marked in blue.

This paper uses real atmospheric observations of $^{14}$C (from flasks) and CO (in situ) that infer ffCO2 values, and convolves them in an atmospheric inversion framework to test the ability of the observations to constrain ffCO2 in an urban area. They demonstrate that even though the $^{14}$C-based ffCO2 observations are more precise, the low sampling density is insufficient to produce robust inversion results. In contrast, they show that CO-based ffCO2 values with larger uncertainties but far denser observations can produce robust inversion results. The paper also acknowledges some of the challenges for urban-scale inversions and presents some interesting variations on the atmospheric inversion framework to deal with these. This includes allowing the inversion to scale only total emissions rather than allowing the inversion to induce spatial variability, and fixing the large point source emissions which are expected to be quite well known.

This is a very nice study and I thoroughly enjoyed reading the paper. I have a few suggestions for clarification in the text, and my main comment is that the Discussion and Conclusion sections are overly long and largely repeat what is already stated in the Results section. I recommend shortening and combining the Discussion and Conclusionsections. I recommend acceptance with these minor revisions.

You are right, we tried to shorten the Discussion and Conclusion sections by combining them in our new manuscript version.

Specific comments:

Language and grammar: Please check throughout for language and grammar. I noticed a number of minor errors that should be corrected.

We went through the text and corrected several language and grammar errors.

Page 5, line 140. As we aim to investigate…. This sentence is confusing and should be rephrased.

Done (p. 6, l. 169ff).

Page 6 Line 155, last sentence of paragraph is confusing. Suggest rephrasing to: Times for the hourly-integrated ΔffCO2 observations are reported as the start of the hour e.g….

Done (p. 6, l. 186ff).

Page 6.  Line 170-174.  Is there a possibility of inducing bias by excluding the two sigma outliers?  It is likely that the outliers represent some specific atmospheric conditions such as low wind speeds, which might also imply cold inversions that could have different emissions than other meteorological conditions.

We think that the 2$\sigma$ filter is a quite soft selection criterion, as it only excludes less than 5% of the data. However, you are right that it excludes more events during winter compared to summer. In the case of the $\Delta$CO-based $\Delta$ffCO$_2$ record, only 18% of the excluded data are from the summer half-year. The remaining 82% of the excluded data are from the winter half-year and especially from synoptic events, which are hard to represent with the model. However, the 2$\sigma$ filtering leads to only two weeks (one week in January and one week in February 2020) with less than 21 hourly afternoon observations per week. Moreover, there are only 21 days within this two-year period from which no observations are used in the inversion. Therefore, we conclude that the potential of inducing a bias with the 2$\sigma$ filter is probably small.

Page 10.  Lines 246-252.  It is not clear from what is written how the authors can be confident that the a-posteriori flux variabilities must indicate the inversion is wrong, versus the priors being wrong.  Please clarify.

We also made an inversion run for which we use the monthly varying emissions from TNO for the area source prior emissions (Fig. 1b below, black dashed line). The a-posteriori flux variability is very similar to the results of the inversion run with the temporally flat prior emissions (Fig. 1a below, black dashed line). From this, we conclude that the large a-posteriori flux variabilities are not caused by wrong, i.e. the temporally flat prior emissions.

[Figure]

*Figure 1: Results of the $^{14}$C-based $\Delta$ffCO$_2$ inversion with (a) flat and (b) monthly varying prior emissions. Fig. 1a is the same as Fig. 3a in the manuscript.*

Page 11.  Lines 293-295.  Does this aggregation of observations apply to the following section 3.2?  Or to the previous section?  Please clarify, and if this

aggregation applies to section 3.2, I suggest moving this text into the start of that section.

*The averaging belongs to the following section 3.2. We shifted this information to that section (p. 14, l. 384ff).*

Page 12.  Lines 303-323.  This section awkwardly splits between two paragraphs of discussion in the main text, and referring to figures that are only presented in the appendix/supplementary material.  I suggest substantially reducing the text in the main document and including the longer discussion in the supplementary material.  Alternatively, move the figures into the main document to match the text (although this will make the paper even longer).

*We substantially reduced the discussion about the appropriate averaging interval in the main text (p. 14, l. 386 – p.15, l. 396) and refer to a more detailed discussion in the Appendix C in the manuscript (p. 26-27).*

Page 13.  Line 356-357.  This sentence needs grammatical correction.

*Done (p. 16, l. 433ff).*

Page 18.  Lines 500-502.  This analysis required guessing what the seasonal cycle might be, which is reasonable for this study.  But would it be realistic to construct a seasonal cycle in the $\Delta CO/\Delta ffCO2$ ratio by estimating the seasonal contributions of each ffCO2 sector and it's characteristic ratio?  I'm not suggesting this needs to be done for this study, but it would be a useful recommendation in the conclusions if indeed it is feasible.

*In the companion paper (Maier et al., 2023a, https://doi.org/10.5194/egusphere-2023-1237) we show that there can be large differences between $^{14}$C-based and inventory-based $\Delta CO/\Delta ffCO_2$ ratios, both at an urban and a remote station. Therefore, we recommend to validate the bottom-up ratios by observations before using them to calculate a continuous $\Delta ffCO_2$ record. We included this information in the manuscript (p. 21, l. 601-605).*

Page 16 – 20.  As noted in my general comments, the discussion and conclusions largely repeat each other, and repeat much of what was said in the results section.  I suggest substantially shortening the discussion section and merging with the conclusions section.

*We tried to shorten the discussion section and merged it with the conclusions section.*

---

## Author Comment (AC2)

We want to thank John Miller for the review of our manuscript and the variety of helpful input and comments to improve this manuscript.

Review of Maier et al, 2023, submitted to ACP, by John Miller

"Potential of 14C-based versus ΔCO-based ΔffCO2 observations to estimate urban fossil fuel CO2 (ffCO2) emissions"

General comments:

This paper presents very promising results showing that continuous CO data, when 'calibrated' with discrete $D^{14}CO_2$ measurements to produce 'pseudo-continuous' fossil $CO_2$ mole fractions (and then averaged over a week to a month) has great potential in estimating urban fossil $CO_2$ emissions. To me, this is the most important result from the study and could be emphasized a bit more. I was impressed by the breadth of sensitivity tests that were conducted, which provide a lot of confidence in the results. The figures were clear (although I have a few small suggestions, and the writing was generally good; I have included some inline comments to help with clarity and English usage.

I am recommending 'accepted subject to minor revisions', because I don't think that at present any single suggestion I'm making is major, but in totality there are a lot of suggested/requested revisions. Additionally, as noted below and in the annotated .pdf I am interested to understand why formal random error was not estimated (or at least presented) in this study. I don't know if incorporating that information would constitute something 'major'.

The main objective of our study was to answer the question of which $\Delta ffCO_2$ information (discrete $^{14}C$-based or continuous $\Delta CO$-based $\Delta ffCO_2$) is best suited to estimate the seasonal cycle of fossil emissions in an urban region. To illustrate the information content of the $\Delta ffCO_2$ observations regarding the seasonal cycle of the $ffCO_2$ emissions, we switched off the seasonal cycle in the a-priori emissions. We could show that in our urban target region only the continuous $\Delta CO$-based $\Delta ffCO_2$ observations (which were calibrated with $^{14}C$-based $\Delta CO/\Delta ffCO_2$ ratios) lead to data-driven $ffCO_2$ emission estimates that are robust enough to be used to validate the seasonal cycles of the emission inventories. We demonstrate the robustness of our results by showing the a-posteriori seasonal cycles for several prior uncertainties (see the "spaghetti" plots in Fig. 3 and 4) and by performing additional sensitivity tests (see Fig. 5 and Fig. C1).

We want to emphasize that our study was not designed to *improve* the emission inventory within the main footprint of Heidelberg. We fully agree, that the calculation of the a-posteriori flux uncertainties would be essential for such a study. However, this would also require a careful estimation of the a-priori flux covariance matrix, thus detailed knowledge about the spatial and temporal correlations of the

prior emissions. However, we do not know these statistics for our regional target domain.

Therefore, we think that the shown spaghetti plots and sensitivity runs are better suited to demonstrate the robustness of the a-posteriori seasonal cycles and thus to answer our main research question than if a-posteriori flux uncertainties are presented, which would depend on a rather subjective choice of the a-priori flux uncertainties.

In addition to my overall positive impression of the paper, I list below numerous general and specific aspects that could (and in some cases need) to be improved.  In no particular order, some general issues:

1. The Discussion section repeats much of what is said just above in the Results section. I personally prefer integrated results + discussion, because I think it is more efficient and (as is the case here) there are inevitably elements of 'discussion' in the results section. Sticking with current format is fine, of course, but the paper would be improved greatly by removing redundancy and focusing the discussion on new ideas and analysis.  As just one example, I would be interested to learn more about what difference between Figs. 3 c and d tell us.  On a related note, it would be interesting to see if you can derive some quantitative results from the large number of sensitivity results run, e.g., sensitivity of posterior to prior uncertainty and some estimate of posterior uncertainty given that this is not otherwise done.

We agree that our Discussion section had much redundancy. We tried to remove redundancy in the manuscript and merged the Discussion and Conclusions section, as was suggested by the other reviewer.

We performed the inversion runs in Fig. 3c and 3d to show that one hypothetical flask per week is not enough to get robust and plausible seasonal cycles in this urban region. The differences between Fig. 3c and 3d indicate that the inversion results are still determined by the model-data mismatch of individual (hypothetical) flasks. This is also illustrated by Fig. 1 (below), which shows the fits of the a-posteriori results to the observations. The inversion mainly reduces the largest model-data mismatches of individual winter flasks. We added this explanation in our manuscript (p. 13, l. 345ff).

[Figure]

*Figure 1. Fits of the flat a-priori and a-posteriori emissions to the observations for (a) hypothetical Tuesday and (b) Friday flasks. These fits correspond to Fig. 3c and 3d in the manuscript.*

2. There is a lot of faith placed in the TNO inventory without having demonstrated (or discussed this). It implicitly serves as a truth metric. I would feel more comfortable with this if you discussed this explicitly saying that you do in fact treat TNO as a truth metric but also saying why it should be treated as such, especially with respect to its seasonality.

Additionally, there are numerous cases where the interpretation of results assumes the point source emissions to be perfectly accurate (all mismatch being assumed to result from transport uncertainty) and also the assumption that the spatial pattern of the area sources is perfectly known. These key assumptions need to be acknowledged more clearly, and the fact that they are not perfect assumptions needs to be recognized in the interpretation of results.

We agree, we placed a lot of faith in the TNO inventory by fixing the point source emissions and by optimizing only one spatial scaling factor for the area source emissions. We now point out potential inaccuracies in the TNO emission inventory (e.g. p. 14, l. 372ff; p.18 l. 495; p. 19 l. 534ff).

Our main purpose of showing the TNO seasonal cycles is to assess whether our a-posteriori seasonal cycles show a plausible amplitude and phasing. However, to get more trust into the TNO emissions, we now compare the TNO emissions in the nearfield of Heidelberg with emissions based on the EDGAR and GridFED inventory (see Fig. G1 in the revised manuscript). We also included a discussion about the differences between the inventories in our manuscript (p. 20, l. 557-569). While the EDGAR emissions are on average about 25% lower than the TNO emissions in the nearfield area of Heidelberg, the GridFED inventory shows ca. 23% larger emissions than TNO. Overall, the seasonal cycles of our top-down estimate are in the range covered by all three bottom-up inventories, thus inferring that we could indeed

reliably reconstruct the amplitude and the phasing of the seasonal cycle from flat a-priori area source $ffCO_2$ emissions with the $\Delta CO$-based $\Delta ffCO_2$ observations.

3. In general, there is a need for more detail to be included in the paper. A thorough list is provided in line as comments in the marked up .pdf, but I'll mention some items here as well.

Thank you for this marked up .pdf. We respond to your comments in the .pdf directly. We tried to implement most of the suggestions.

a. I think it's important to quantify how different versions of the inverse model fit the observations. Reduced chi-squared, std. dev, and mean bias (please separate sd and bias instead of using RMSE which blends these) are important and easy to calculate metrics. These are especially important when trying to demonstrate things like overfitting. On a related note, it doesn't appear to be the case, but were any observations withheld for cross validation?

Thank you for these suggestions. We included fits of the a-posteriori results to the observations (see Fig. B1, Fig. C3 in the manuscript), and calculated mean biases and standard deviations (e.g. p. 12, l. 316ff; p.17, l. 450ff; p. 27, l. 768ff). For example, Fig. B1 illustrates the fitting of individual flasks with large model-data mismatches. For the $^{14}C$-based $\Delta ffCO_2$ inversion, we also performed a reduced chi-squared analysis (p. 12, l. 320-324). However, the reduced chi-squared values range from 1.10 (for a prior uncertainty of 50%) to 0.97 (for a prior uncertainty of 150%), which already lead to large and unrealistic variabilities in the a-posteriori seasonal cycles. Therefore, we conclude that in our case the reduced chi-squared analysis might be less suitable to show overfitting.

b. A brief mention of how the non-fossil parts of the radiocarbon budget are treated in the construction of atmospheric CO2ff, especially the nuclear reactor flux of $^{14}CO_2$, would be useful.

We decided to not include this detailed information, as we extensively describe the construction of the $^{14}C$- and $\Delta CO$-based $\Delta ffCO_2$ concentrations and their uncertainties in the companion paper (Maier et al., 2023a, https://doi.org/10.5194/egusphere-2023-1237). We set up the companion paper with the intention to provide the observational basis for the present manuscript, which then can fully focus on the inversion.

c. The inversion methodology deserves some description, mainly basic aspects such as that it is not an 'analytical' inversion (i.e. an exact solution to the cost function minimum).

We included a more detailed description of the inversion methodology in the Appendix A in the manuscript (p. 23).

Given the small size of the state vector (the discretization of which would be good to explicitly describe) I would expect an analytical solution would be entirely possible just via a basic matrix inversion. Is there a reason this approach was not used and the more complicated R2005 approach was? The main significance is that a fully accurate posterior covariance could have been calculated allowing for presentation of analytically exact random error and also estimation of degrees of freedom, correlations over time, etc. And even in R2005 (according to my reading of it) a reasonable posterior covariance approximation should be available but none of these results are presented.

While the sensitivity tests address quite a few systematic error issues, it's unclear why the random errors were not presented.

We agree, the small size of the state vector might allow an analytical solution via matrix inversion. However, we decided to use the CarboScope framework instead due to two reasons: (1) The conjugate gradient algorithm used in the CarboScope framework leads also to a fast convergence because of the small state vector. (2) Using the CarboScope framework would allow the implementation of larger state spaces (e.g. to investigate in a next step the emissions from individual ffCO$_2$ sectors like heating or traffic without the need to set up a new inversion system). And this would avoid to set up two different inversion systems.

Regarding the a-posteriori random errors, please refer to our first answer above.

4. While a small point, I think it's important to clarify that the Delta(CO)-based method is actually based on both CO and 14C. Without this, readers may think CO and CO$_2$ alone have the capability to constrain fossil CO$_2$.

We fully agree, this may lead to confusion. We tried to make this clearer, also in the abstract (see p. 1, l. 21ff; p. 3, l. 77ff).

Specific comments are embedded as comments inline in the .pdf.

See our answers to your specific comments in the attached .pdf.

---

## Author Comment (AC3)

[revised manuscript text omitted]